# Are sEMG, Velocity and Power Influenced by Athletes’ Fixation in Paralympic Powerlifting?

**DOI:** 10.3390/ijerph19074127

**Published:** 2022-03-31

**Authors:** Ialuska Guerra, Felipe J. Aidar, Gianpiero Greco, Paulo Francisco de Almeida-Neto, Michele De Candia, Breno Guilherme de Araújo Tinoco Cabral, Luca Poli, Mauro Mazini Filho, Roberto Carvutto, Ana Filipa Silva, Filipe Manuel Clemente, Georgian Badicu, Stefania Cataldi, Francesco Fischetti

**Affiliations:** 1Federal Institute of Education, Science and Technology of Ceará (IFCE), Campus of Juazeiro do Norte, Juazeiro do Norte 63040-540, Brazil; ialuskaguerra@gmail.com; 2Department of Physical Education, Federal University of Sergipe (UFS), Saint Cristopher 49100-000, Brazil; fjaidar@gmail.com; 3Group of Studies and Research of Performance, Sport, Health and Paralympic Sports (GEPEPS), Federal University of Sergipe (UFS), Saint Cristopher 49100-000, Brazil; 4Graduate Program of Physiological Science, Federal University of Sergipe (UFS), Saint Cristopher 49100-000, Brazil; 5Graduate Program of Physical Education, Federal University of Sergipe (UFS), Saint Cristopher 49100-000, Brazil; 6Department of Basic Medical Sciences, Neuroscience and Sense Organs, University of Study of Bari, 70124 Bari, Italy; gianpiero.greco@uniba.it (G.G.); michele.decandia@uniba.it (M.D.C.); luca.poli@uniba.it (L.P.); roberto.carvutto@uniba.it (R.C.); francesco.fischetti@uniba.it (F.F.); 7Department of Physical Education, Federal University of Rio Grande do Norte (UFRN), Natal 59078-970, Brazil; paulo220911@hotmail.com (P.F.d.A.-N.); brenotcabral@gmail.com (B.G.d.A.T.C.); 8Graduate Program of the School of Physical Education and Sports, Federal University of Juiz de Fora, Juiz de Fora 36036-900, Brazil; personalmau@hotmail.com; 9Escola Superior Desporto e Lazer, Instituto Politécnico de Viana do Castelo, Rua Escola Industrial e Comercial de Nun’Álvares, 4900-347 Viana do Castelo, Portugal; anafilsilva@gmail.com (A.F.S.); filipe.clemente5@gmail.com (F.M.C.); 10Research Center in Sports Performance, Recreation, Innovation and Technology (SPRINT), 4960-320 Melgaço, Portugal; 11The Research Centre in Sports Sciences, Health Sciences and Human Development (CIDESD), 5001-801 Vila Real, Portugal; 12Instituto de Telecomunicações, Delegação da Covilhã, 1049-001 Lisboa, Portugal; 13Department of Physical Education and Special Motricity, Transilvania University of Brasov, 500068 Brasov, Romania; georgian.badicu@unitbv.ro

**Keywords:** muscle activation, dynamic force, bench belts, bench press, parapowerlifting

## Abstract

The bench press is performed in parapowerlifting with the back, shoulders, buttocks, legs and heels extended over the bench, and the use of straps to secure the athlete to the bench is optional. Thus, the study evaluated muscle activation, surface electromyography (sEMG), maximum velocity (MaxV) and mean propulsive velocity (MPV), and power in paralympic powerlifting athletes under conditions tied or untied to the bench. Fifteen experienced Paralympic powerlifting male athletes (22.27 ± 10.30 years, 78.5 ± 21.6 kg) took part in the research. The sEMG measurement was performed in the sternal portion of the pectoralis major (PMES), anterior deltoid (AD), long head of the triceps brachii (TRI) and clavicular portion of the pectoralis major (PMCL). The MaxV, MPV and power were evaluated using an encoder. Loads of 40%, 60%, 80% and 100% 1RM were analyzed under untied and tied conditions. No differences were found in muscle activation between the tied and untied conditions; however, sEMG showed differences in the untied condition between AD and TRI (F (3112) = 4.484; *p* = 0.005) in the 100% 1RM load, between PMCL and AD (F (3112) = 3.743; *p* = 0.013) in 60% 1RM load and in the tied condition, between the PMES and the AD (F (3112) = 4.067; *p* = 0.009). There were differences in MaxV (F (3112) = 213.3; *p* < 0.001), and MPV (F (3112) = 248.2; *p* < 0.001), between all loads in the tied and untied condition. In power, the load of 100% 1RM differed from all other relative loads (F (3112) = 36.54; *p* < 0.001) in both conditions. The tied condition seems to favor muscle activation, sEMG, and velocity over the untied condition.

## 1. Introduction

In parapowerlifting competitions, the bench press occurs with the athlete’s body in a supine position on a bench. The head, shoulders, buttocks, legs buttocks, legs (as extended as possible) heels (if any) must be on the bench from start to finish of the lift; in addition, it is advised that the lower limbs be tied to the seat [1,2]. 

In this way, the execution of the bench press movement consists of the following phases: The initial phase is marked by the removal of the bar from the rack and positioning with the arms extended and elbows locked. The eccentric phase is characterized by the movement of the bar down until it touches the chest or abdominal area. Then, in the concentric phase, the bar is pushed up until, at the end of the lift, the arms are extended, and the elbows are locked again. The athlete must remain in this position until the referee authorizes the return of the bar in the rack [1,3]. 

During all phases of the lifting, the adopted body position must be maintained, allowing the athlete to use up to two leg/bench straps to tie his legs to the bench [1,2]. The goal of tying your legs to the bench is to improve body stability, which has been shown to be a possible influence on performance during bench press in a powerlifting competition [4,5]. 

In addition, in this sport, the arch bridge technique (i.e., the athlete performs a marked hyper lordosis in the spine, with scapular retraction, to perform the movement) is preferably adopted by the athletes, the referred technique loses efficiency because it is performed with the legs extended and tied over the bench, this implies a low transfer of force for lifting the loads [4,5]. In this sense, the lashing of the legs with straps can be a limiting factor of the specific performance in Paralympic powerlifting.

The bench press is characterized by muscle activation, dynamic force variables, and speed of movement of the bar in the concentric phase of the movement, with the occurrence of variations in muscle activation as a result of the body position adopted at the time of lifting [6]. The lifting of the bar performed with maximum speed is a presupposition for the acquisition of dynamic force variables [7]. Composing this scenario, morphophysiological variables are associated with the development of strength, especially in the upper body, and muscle mass, fiber types and muscle architecture stand out in this context [7].

Therefore, this study hypothesizes that compared to using the free legs, performing the bench press in paralympic powerlifting with the legs tied will promote less muscle activation limiting the execution velocity and propulsive strength of the athletes. Thus, this research aimed to analyze the variations in sEMG, maximum velocity, mean propulsive velocity and muscle power of athletes of paralympic powerlifting during the execution of bench press with various loads (40%, 60%, 80% and 100% 1RM) performed with legs tied and untied on the bench.

## 2. Materials and Methods

### 2.1. Experimental Design

The study was developed at the Federal University of Sergipe—SE/Brazil, over a period of 3 weeks. On the first day, the athlete was weighed and the test of 1 repetition maximum took place. Then, electromyographic data were collected during the bench press under untied and tied conditions with a leg/bench straps with loads of 40%, 60%, 80% and 100% of 1RM. Athletes were randomized considering the decreasing order of the 1RM percentage values of each athlete. Figure 1 illustrates the experimental design of the research.

### 2.2. Sample

Fifteen Paralympic powerlifting athletes, with various disabilities and considered elite athletes, participated in the study, participating in competitions certified by the Brazilian Paralympic Committee and, therefore, eligible for the sport [1]. Table 1 presents the sample characterization.

Athletes voluntarily participated in the research by signing an informed consent form in accordance with Resolution 466/2012 of the National Research Ethics Commission (CONEP) of the National Health Council and the ethical principles Declaration of Helsinki. The experimental design was submitted (CAEE ID: 79909917.0.0000.55.46) and approved by the Ethics Committee in Research with Human Beings of the Federal University of Sergipe (UFS), under Statement Number 2,637,882.

### 2.3. Intervention

The intervention process started by warming up the upper limbs through abduction and rotation of the shoulders using dumbbells and extension of the elbows on the pulley. Each exercise was performed three times with 10 to 20 repetitions. Next, the athletes performed a set of bench press with 10 lifts at a slow speed ratio (3:1 s, eccentric: concentric) and 10 more fast speed lifts (1:1 s, eccentric x concentric), both with load of 30% 1RM. To complete the warm-up on the bench press, five sets of five lifts were performed with a fixed load of 85 to 90% 1RM and a rest interval of 3–5 min. During the test, athletes received verbal stimulation to achieve maximum performance [9,10] but no type of feedback was applied on the lift performed. The lifts were performed on a specific bench press for parapowerlifting (Eleiko Sport AB, Halmstad, Sweden), approved by the International Paralympic Committee [1] with a total length of 210 cm. The bar used in the lifts is 2.2 m long, weighs 20 kg, is serrated, has grooves and a marking to delimit the athletes’ footprint (between 42 cm and 81 cm), according to official IPC rules [1].

### 2.4. Body Weight

The athletes’ body weight was measured on a digital platform-type electronic scale (Micheletti^®^, São Paulo, SP, Brazil), with a maximum capacity of 300 kg and dimensions of 1.50 × 1.50 m [11,12].

### 2.5. Surface Electromyography—sEMG

Muscle activation was measured in the triceps brachii (long head), in the anterior deltoid and in the pectoralis major–sternal and clavicular portion. The electromyographic signals were captured using through double type electrodes positioned on the right side of the body in parallel to the muscle fibers, located 2 cm from the center at the point of the area of greatest muscle amplitude. The ground electrode was placed over the olecranon [2,13,14]. The area of skin used for electrode placement was shaved and cleaned with alcohol before the electrode fixation procedure. To acquire the signals, the athletes performed 1 series with 1 repetition in each of the proposed conditions (tied × untied) with a 4 min rest break between each lift.

Data were captured by an electromyograph (MIOTEC^®^, Porto Alegre, RS, Brazil), with an 8-channel input. To filter the signals, a second order Butterworth bandpass filter of 20–500 Hz and a notch of 60H was used. To calculate the signal amplitude, the root mean square (RMS) was used, with a window of 100 RMS, with normalization by percentage of the maximum voluntary isometric contraction. (CVMI), considering the time recorded by the equipment program in which each athlete used to perform the concentric phase of the movement. The signal normalization process began before the test was carried out with the determination of the CVMI through the execution of an isometric bench press lifting with a duration of 6 s. The CVMI values obtained were recorded in the electromyography software and used for normalization. The normalized values can be accessed in the report provided by the equipment software and were used for analysis in this study [9,10,14].

### 2.6. Dynamic Force Variables

Maximum velocity (MaxV), mean propulsive velocity (MPV) and power were calculated by a Linear Encoder (Force Measurement System Vitruve; Mostoles, Madrid, Spain) [15]. The Encoder measured the parameters of vertical displacement velocity, being evaluated VMax, MPV and power from the beginning of the concentric phase until the moment when the athletes’ elbows were fully extended [2]. This equipment showed reliability during the bench press [16].

### 2.7. Maximum Load Test (1RM)

The 1RM test was used to determine the load to be lifted by athletes in the bench press on an official parapowerlifting bench [1]. The initial attempt was performed with a self-selected load to be lifted by the athlete with maximum effort only once. From this first load, weight increments were performed up to the limit in which the athlete could perform only one repetition. If the athlete was unable to complete a lift, the load was reduced by 2.4 to 2.5% and a new attempt would be made. Between each attempt, rest intervals of 3 to 5 min were performed. The 1RM test was performed during the familiarization period specified in the experimental design [17,18].

### 2.8. Statistical Procedures

In this study, descriptive statistics were used with measures of central tendency, mean (X) ± standard deviation (SD). The normality of the variables was verified using the Shapiro–Wilk test, in accordance with the sample size. For comparisons between groups, ANOVA (two Way) (Muscle × condition—tied and untied), with post hoc Bonferroni test was carried out. In order to estimate the effect size for between-lift comparison Cohen’s d was calculated as the difference between the mean divided by the pooled SD [19]. Low effect values (≤0.05), medium effect (0.05 to 0.25), high effect (0.25 to 0.50) and very high effect (>0.50) were considered as parameters to stipulate the effect size (partial Eta squared: η2p), for ANOVA [19]. To verify the reliability of the measurements, the intra-class correlation coefficient was used. The statistical procedures were carried out in the Statistical Package for the Social Science (SPSS) (Armonk, NY, USA: IBM Corporation), version 22.0. The level of significance was set at *p* < 0.05.

## 3. Results

### 3.1. Surface Electromyography—EMG

The bench press performed with a load of 100% of 1RM, in the condition tied with a leg/bench straps, presents, in the concentric phase, higher values of muscle activation in the triceps brachii and in the pectoralis major–clavicular portion. In the untied condition, the highest values occurred in the triceps brachii and in the pectoralis major–sternal portion. The two-way ANOVA showed no significant differences in muscle activation between the tied and untied conditions but indicated ((F (3112) = 4.444; *p* = 0.005)) between the anterior deltoid and the triceps brachii in the untied condition (Figure 2). The Intraclass Correlation Coefficient showed reliability between measurements in this % 1RM (CCI = 0.726; *p* < 0.001).

With regard to the 80% 1RM load, the bench press performed in the concentric phase, presented, in both conditions (tied and untied), higher values of muscle activation in the pectoralis major–clavicular portion and in the triceps brachii. Two-way ANOVA did not indicate significant differences in muscle activation between the analyzed muscles or between the tied and untied conditions. The Intraclass Correlation Coefficient showed reliability between measurements in this % 1RM (CCI = 0.652; *p* = 0.029).

At the 60% 1RM load, the pectoralis major–clavicular portion and in the triceps brachii showed higher values in the tied condition (PMCL = 149.0 + 111.0; TB = 123.0 + 104.0) while in the untied condition the muscle activation occurred in the pectoralis major in the clavicular portion and also in the sternal portion (PMCL = 253.0 + 252.0; PMES = 227.0 + 197.0). Two-way ANOVA did not indicate significant differences in muscle activation between the two conditions analyzed but detected differences (F (3112) = 3.743; *p* = 0.013) in the untied condition between the pectoralis major–clavicular portion and two other muscles (Figure 3). The Intraclass Correlation Coefficient showed reliability between measurements in this % 1RM (CCI = 0.668; *p* = 0.001).

Regarding the 40% RM load, muscle activation showed higher values in the pectoralis major–sternal portion and in the triceps brachii in the tied and also untied condition. Two-way ANOVA did not indicate significant differences between the conditions under study, but rather, (F (3112) = 4.067; *p* = 0.009) between the pectoralis major and the anterior deltoid in the tied condition (Figure 4). The Intraclass Correlation Coefficient showed reliability between measurements in this % 1RM (CCI = 0.751; *p* < 0.001).

### 3.2. Velocity and Power

The velocity and power variables were measured in bench press lifts at relative loads (% 1RM) from 40% to 100%, in conditions untied and tied by a leg/bench strap. The data indicates a reduction in the maximum velocity and mean propulsive velocity values as the relative load increases. Regarding power, it is possible to observe the same behavior of this variable, except for the relative load of 40% 1RM. The values of the velocity variables are detailed in Table 2.

Significant differences were found (F (3112) = 213.3; *p* < 0.001) in the maximum velocity of bench press lifts between all loads (% 1RM) in the tied (a,b,c,d) and untied (e,f,g,h) conditions. The Intraclass Correlation Coefficient showed reliability between measurements in MaxV (CCI = 0.815; *p* < 0.001).

The same occurred in the mean of propulsive velocity (F (3112) = 248.2; *p* < 0.001). As for power, the 100% 1RM load differed from all other relative loads (F (3112) = 36.54; *p* < 0.001) in both bench press conditions (Figure 5). The Intraclass Correlation Coefficient showed reliability between measurements in MPV (CCI = 0.921; *p* < 0.001).

Strong negative correlations between the relative load (% 1RM), maximum velocity and mean propulsive velocity were observed in the untied conditions (r = −0.984, ICC 95% −0.999 to −0.442, r^2^ = 0.969; r = −0.991, ICC 0.999 to −0.649, r^2^ = 0.983), and tied (r = −0.998, ICC −1000 to −0.929), r^2^ = 0.997, r = −0.999; ICC −1.000 to −0.931; r^2^ = 0.997), indicating a reduction in the values of velocity variables due to the increase in the relative load in bench press lifts.

## 4. Discussion

The aim of this study was to analyze the variations in muscle activation, and in the variables of velocity (maximum velocity and mean propulsive velocity) and power, in bench presses with relative loads (% 1RM) comparing the data in conditions untied and tied with leg/bench straps. Muscle activation allows us to estimate which musculature is most requested during an exercise, that is, which are the main muscles involved in the kinetics of a given movement [9,20].

Thus, it was verified in this study if the use of leg/bench straps that hold the lower limbs of paralympic athletes to the bench press would promote some alteration of the musculature used to perform the lifting in paralympic powerlifting and what the behavior of this variable is when implementing different relative loads (% 1RM) [4,5].

The results show that the predominance of activation of the pectoralis major clavicular portion in the tied condition and pectoralis major sternal portion in the untied condition in loads of 80% to 40% 1RM. In the same direction [21], with athletes of paralympic powerlifting, also showed higher muscle activation values in the pectoralis major clavicular portion and pectoralis major sternal portion. Similarly, the study [22] also indicated greater activation of the pectoralis major during the bench press exercise performed by athletes of the paralympic powerlifting.

Only the occurrence, in our study, of higher values of muscle activation in the triceps brachii at a relative load of 100% 1RM, differs from the trend found in these other two studies. Elevated muscle activation of the triceps may occur due to fatigue of the pectoral muscles [23] which makes the triceps a critical position for performance in Paralympic weightlifting.

On the other hand, in a systematic review carried out by Stastny [24], which analyzed 14 studies on electromyographic analysis in bench press lifts, considered the brachial triceps as the musculature most sensitive to changes in muscle activation, and may present different values according to the intensity of the exercise and movement velocity, among other factors.

The non-occurrence of significant differences in muscle activation between the two conditions tied vs. untied suggests the need to consider the degree of motor control of the trunk, in order to infer the degree of influence of the use of leg/bench straps on performance variables in the bench press in paralympic powerlifting athletes because as Pérez-Trejos [25] points out, the muscular condition of the trunk is essential for maintaining posture during sports practice. Muscle activation was modified according to the load lifted by each athlete during the test (40% 1RM, 60% 1RM, 80% 1RM e 100% 1RM). The dispersion of sEMG values was also identified in the study by Aedo-Muñoz [6] when evaluating Paralympic weightlifting athletes.

Regarding the variables of maximum velocity and mean propulsive velocity, it is noticeable a greater slowness in carrying out the lifting of the bar in the higher relative loads. The same trend occurred in the study [26] when comparing the velocity–load during bench press relationship in athletes performing the movement with the back in a natural lumbar arch and moderate scapular retraction vs. the back with a lumbar arch pronounced and scapular retraction and, in the study [27] when analyzing the velocity of displacement of the bar in the bench press in a tied vs. untied condition.

A strong load–velocity relationship was found in both lift conditions. In the same direction, the study [28], highlights strong correlations between the bar velocity and the relative load, especially in loads from 70% 1RM and Martínez-Cava et al. [29], also points out reductions in values in the bar velocity during the bench press.

Regarding power, there was a greater reduction in the relative load of 100% 1RM, although the data did not indicate a strong load-to-power ratio. A study about Changes in Bench Press Velocity and Power [30], in turn, indicates a reduction in power values, as the relative load increases.

The current results revealed the importance of choosing whether or not to use leg/bench straps to favor performance in paralympic powerlifting, considering increasing the velocity of movement and the production of strength in a shorter time, with greater activation of the primary motor muscles. This highlights the possibility of providing, in addition to the athlete’s free choice, whether or not there is a need to use this material in order to improve the athlete’s performance in competitions and also to avoid possible injuries. Thus, it becomes important to include the alternation between the use and non-use of bench belts during training until the occurrence or not of benefits for the athlete’s performance becomes clear.

The parapowerlifting rules [7] considers the use of bench straps optional because it considers the needs and specificities of each type of disability, and although the present study did not show significant differences in muscle activation, such differences were found in maximum velocity indicating the possibility of benefiting from the use of the bench straps. Thus, if, the use of bank lanes can be considered a limiting factor [4,5] on the other hand, it can act as an element that facilitates performance. In addition, research on the variables addressed, the kind of measurements and the level of the athletes are strong points in this study and become relevant aspects for training and for the outcome of competitions.

However, despite the relevant results, our study has the following limitations: the reduced number of athletes in each specific type of disability and in each sports category did not allow us to clarify in which specificities it would be more appropriate to use the leg/bench straps to favor the performance of the power. It is also necessary to understand the possible influences of upper body lean mass evaluation, fiber type composition, muscle architecture characteristics and perhaps gender differences in the paralympic bench press. Finally, it would be necessary to consider the possibility of carrying out another study using a counterbalanced design and providing simultaneous feedback and verbal encouragement during the survey in order to verify the influence of these variables on the study.

In addition, it is necessary to consider the behavior of the variables in the conditions analyzed also in the eccentric phase of the bench press movement and especially in the stick point region in order to obtain a more detailed understanding of the use of the leg/bench straps in paralympic powerlifting. In this sense, it is also pertinent to carry out other studies that combine other morphological, biodynamic, and functional variables that may impact the performance of paralympic weightlifting athletes.

## 5. Conclusions

In conclusion, variations in muscle activation occur similarly in untied and tied conditions with a seat belt with a predominance of the pectoralis major clavicular and sternal portion, followed by the triceps brachii. In addition, it is possible to consider that the occurrence of the relationship between the load and the velocity in the displacement of the bar can favor the load-power relationship, especially in the tied condition. Thus, the performance in the Paralympic bench press seems to be more effective when using the tied condition.

## Figures and Tables

**Figure 1 ijerph-19-04127-f001:**
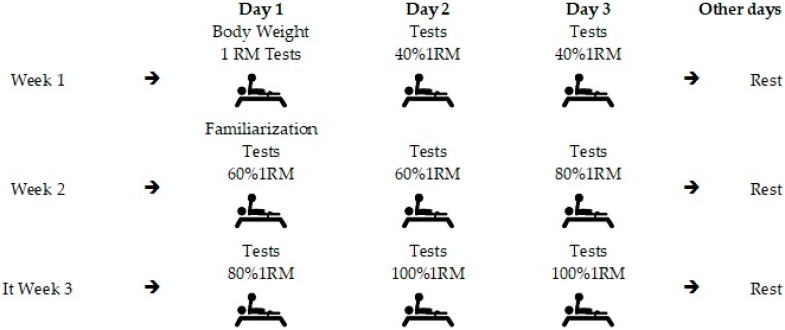
Experimental sequence—planning weekly tests. The tests were in relation to sEMG, MaxV, MPV and power.

**Figure 2 ijerph-19-04127-f002:**
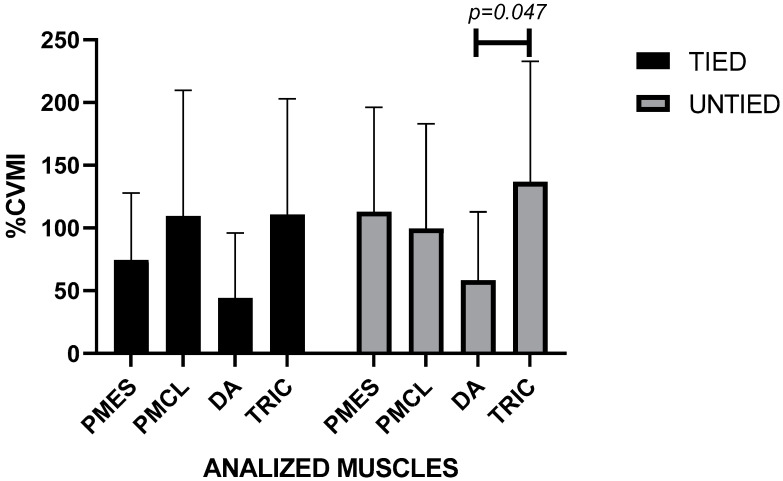
Muscle activation with a relative load of 100% 1RM in conditions untied and tied with leg/bench straps in paralympic powerlifting athletes. Legend: PMES = Pectoralis Major sternal portion; PMCL = Pectoralis Major clavicular portion; DA = Anterior deltoid; TRIC = triceps brachii; CI = −155.9 to 0.675.

**Figure 3 ijerph-19-04127-f003:**
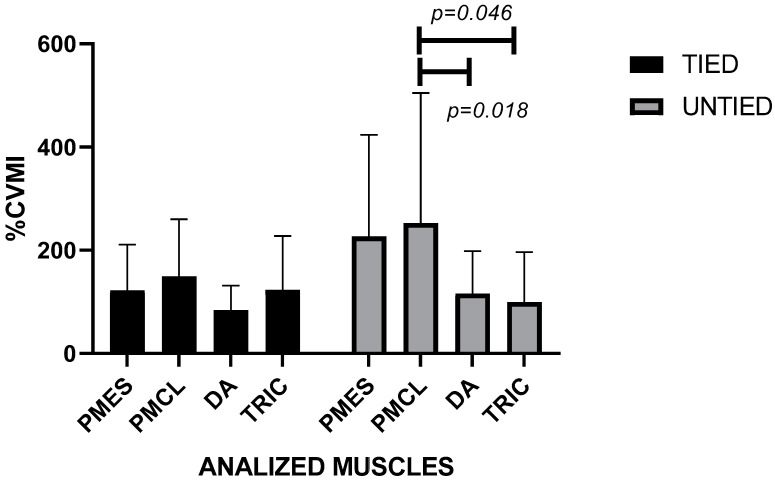
Muscle activation with a relative load of 60% 1RM in conditions untied and tied with leg/bench straps in paralympic powerlifting athletes. Legend: PMES = Pectoralis Major sternal portion; PMCL = Pectoralis Major clavicular portion; DA = Anterior deltoid; TRIC = triceps brachii.

**Figure 4 ijerph-19-04127-f004:**
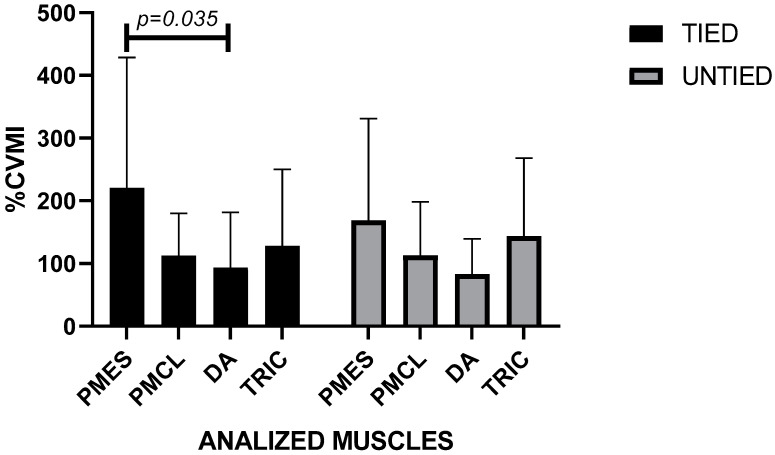
Muscle activation with a relative load of 40% 1RM in conditions untied and tied with leg/bench straps in paralympic powerlifting athletes. Legend: PMES = Pectoralis Major sternal portion; PMCL = Pectoralis Major clavicular portion; DA = Anterior deltoid; TRIC = triceps brachii.

**Figure 5 ijerph-19-04127-f005:**
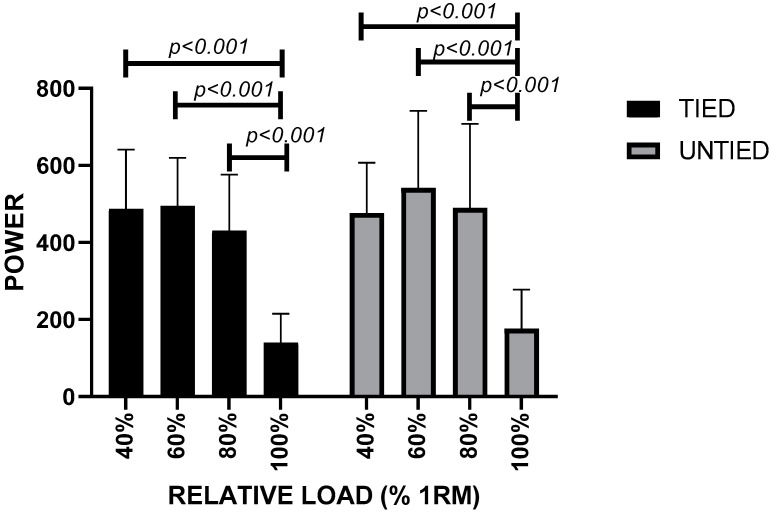
Power vs. relative load during bench press in untied and tied conditions with leg/bench straps in paralympic powerlifting athletes.

**Table 1 ijerph-19-04127-t001:** Sample characterization.

Variables	(Mean ± SD)
Age (years)	22.27 ± 10.30
Weight (Kg)	78.50 ± 21.67
1RM Adapted Bench press (Kg)	114.00 ± 37.19
1RM/weight	1.5 ± 0.46 **

All athletes perform lifts with loads that place them in the top ten of their categories in the country, ** Athletes with lifts above 1.4 on the Bench Press (1-RM/Body Weight) can be classified in the elite category, according to Ball & Wedman [8]. Legend: 1RM: One Repetition Maximum.

**Table 2 ijerph-19-04127-t002:** Comparison between the velocity variables obtained in relative loads (% 1RM) during bench press lifting in untied conditions and tied with a leg/bench straps in powerlifting athletes.

LOAD	BP VARIANT	MaxV (M/S)	MPV (M/S)	POT
40% RM	Tied	1.6 ± 0.18 ^(a,b,c,d)^	1.12 ± 0.16 ^(a,b,c,d)^	487.0 ± 154.0 ^(d)^
	Untied	1.5 ± 0.18 ^(e,f,g,h)^	1.09 ± 0.14 ^(e,f,g,h)^	476.0 ± 131.0 ^(g)^
60% RM	Tied	1.1 ± 0.28 ^(a,b,c,d)^	0.76 ± 0.18 ^(a,b,c,d)^	495.0 ± 124.0 ^(d)^
	Untied	1.24 ± 0.20 ^(e,f,g,h)^	0.85 ± 0.14 ^(e,f,g,h)^	542.0 ± 200.0 ^(g)^
80% RM	Tied	0.76 ± 0.16 ^(a,b,c,d)^	0.50 ± 0.09 ^(a,b,c,d)^	431.0 ± 146.0 ^(d)^
	Untied	0.89 ± 0.28 ^(e,f,g,h)^	0.59 ± 0.19 ^(e,f,g,h)^	490.0 ± 218.0 ^(g)^
100% RM	Tied	0.28 ± 0.09 ^(a,b,c,d)^	0.14 ± 0.07 ^(a,b,c,d)^	140.0 ± 74.7 ^(d)^
	Untied	0.33 ± 0.14 ^(e,f,g,h)^	0.17 ± 0.10 ^(e,f,g,h)^	176.0 ± 101.0 ^(g)^

Legend: BP, bench press; MaxV, maximum velocity; MPV, mean propulsive velocity; POT, power. The results represent Mean ± SD (X ± SD). ^(a,b,c,d)^ = *p* < 0.001.

## Data Availability

The data presented in this study are available on request from the first author.

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
