# Peer review of "Are sEMG, Velocity and Power Influenced by Athletes’ Fixation in Paralympic Powerlifting?"

_ijerph, 2022, doi:10.3390/ijerph19074127_

Round 1

Reviewer 1 Report

This work is a very interesting and elaborate study. The question posed has a high practical relevance.

Abstract

The abstract is written clearly and concisely. It would be helpful, however, to state the results specifically with reference to the central question. After all, it was about the comparison between tied vs untied.

methodology

Section 2.5 describes the technical implementation of dynamic force variables. A statement on the reliability would be necessary here. Because it is not clear if such a device is used whether it also measures reliably.

Section 2.6: it is called root mean square (RMS), what time window was chosen for smoothing, what number of repetitions was used for the EMG analysis?

Section 2.8: was variance homogeneity tested? In the results section, correlation analysis/reliability analyses were calculated using ICC. However, this is not mentioned here. It is unclear why this was done, as it was not part of the question.

Results

Fig 2: Why is the confidence interval given in the figure? This is not in the rest of the figure, if it is necessary then it should be placed in the figure caption.

Table 2: for a better overview I recommend to mark significant differences with symbols (e.g. 100% RM MaxV).

Discussion

Seems a bit superficial and general. The quality of the EMG measurement should be dealt with more intensively. The data clearly show a high dispersion of the EMG parameters between the test persons. This point has certainly calculated one or the other difference between the conditions as not significant. Why do the athletes show such different muscle activations between each other? There are numerous studies that have looked at experts and novices in terms of muscle activation, including the bench press, novices tend to show higher levels of muscle activation. This should be addressed. Not all subjects will benefit equally in terms of performance and muscle activation. Some may also feel that their performance is hindered by tied. There is no mention at all of the practical training measures that could be taken. The reference to the fact that the results could be interesting for trainers is unnecessary.

Reviewer 2 Report

Are Activation, Velocity and Power Influenced by Athletes’ Fixation or Not in Paralympic Powerlifting?

The study aimed to investigate whether a tied or untied condition in bench press enhance muscle activation, barbell velocity and power output in Paralympic power-lifters. I comment the authors for the research question and the effort to conduct this study. As a fan of strength sports I really enjoy reading the manuscript although, as strength and conditioning researcher I have some major and minor comments and suggestions. Bellow I provide my main concerns and following I present my point by point comments and suggestions.

General comments:

  1. I am not sure I found that inside the methods but, was there a counterbalanced design in the experimental procedures? How athletes were randomized during measurements?

  1. Why the actual performance (bench press in kg) was not included in the two different conditions (tied and untied)?

  1. What was the role of upper body lean mass on performance? This should be discussed.

Specific comments:

Title:

A small suggestion here: Reading the word “activation” someone might think that this paper might refer to the PAPE phenomenon. However, first lines of abstract clarify that this activation refers to “muscle activation” though sEMG. I suggest to Author considering changing this word in the title and connect it with the sEMG used in the study.   

Abstract:

Abstract is in a very good state and provides a good overview of the study. No comments here. However, I suggest to Authors to use different key-words not included in the title.

Introduction:

Introduction is well written but needs more strength. Although Authors provides details for the bench press procedure and technique there is a lack of scientific data regarding the muscle activation during bench press [ie. Krzysztofik et al., 2021, Journal of Electromyography and Kinesiology, 56, 102513; Stastny et al., 2017, PLoS ONE, 12(2)] and the analytically presentation of the importance of tied and untied conditions in bench press performance. Thus, Authors have to connect the dots between practise and theory and maybe use references from studies with healthy participants.

Also, there are many parameters that might influence bench press performance such as muscle mass, lean mass, fiber type composition, etc. Please, add a few lines about these factors.

Line 75: Change the phrase “lifting the bench press’ to “lifting the load”.  

Materials and Methods:

Methods are in a good state. There are a few points here that need improvement.

Experimental design: How the athletes randomized during the weekly test schedule? Was there a counterbalanced design applied in the study? This is a major methodological issue that Authors have to clarify in experimental procedures.

Figure 1 is perfect. If a counterbalanced design was used, please add some details in the figure as well.

Sample: Question here: Did all the athletes have the same motor difficulties? Also, since the value 1RM/weight is above 1.4 then why not refer to these athletes as elite Paralympic weightlifters inside the manuscript? This might be a strong aspect of the study.

2.2 Body weight: This paragraph focus on the body mass measurement of athletes. Why Authors add the instruments of bench press measurement? I suggest changing the paragraph title and include both descriptions.

I suggest a small change in the order of paragraphs. Please transfer the paragraph 2.7 “Intervention” under the Sample paragraph. This will be better for readers to closely read the study design and training intervention. Then proceed with the paragraph “body weight”.

2.5. Dynamic Force Variables: Was there a simultaneous feedback from the mean velocity during the tests? This might affect the results of the study since feedback seems to favour performance (Weakley et al., 2021; Velocity-Based Training: From Theory to ApplicationStrength and Conditioning Journal).

Line 132: Correct the abbreviation to MaxV.

Question: Regardless the verbal stimulation to achieve maximum performance, did the Authors gave the instruction to push the load as fast as possible regardless of the actual movement velocity (Blazevich et al., 2020; Sports Medicine, 50, 943–963)?

2.8. Statistical Procedures: Please add intra class correlation coefficients for all measurements to ensure the reliability of measurements.

Question: Why the actual performance in bench press was not measured during the tied and untied condition?

Results:

I suggest to Authors to present the results in the same order the measurements were presented inside the methods.

Line 194: Please delete the phrase “these same muscles” and replace it with “pectoralis major - clavicular portion and in the triceps brachii”.

Why it is important to compare the muscle activation during bench press between the same conditions?

Table 2: If there are significant differences between conditions and different velocities please add marks inside the Table 2. Again, from the mean velocity between tied and untied conditions it seems to me that during the untied condition athletes may lift a greater load.

Figures are very good. Well done. I missed the figure with the relative load of 80% 1RM.

Discussion:

Lines 259-260: When pectoral muscles are fatigued during the bench press, the activation of triceps brachii is increased implying that the triceps brachii may receive a strong hypertrophic stimulus with the continuing number of repetitions (Brennecke et al., 2009; The Journal of Strength and Conditioning Research, 23(7), 1933–1940). This might partially explain the difference in triceps activation during the 100% load condition.

Line 285: Please rephrase “the other study”.

I am following the sceptical of Authors inside the discussion but I miss the answer to the hypothesis from the introduction and a strong take home message for athletes and coaches. Should they use the tied condition or the untied condition?

Limitations also have to include the lack of upper body lean mass evaluation, fiber type composition, muscle architecture characteristics and perhaps gender differences. Also, authors have to add the strong points of the study which might include the level and the number of the athletes, the measurements etc.

This is a very good, organized, scientific study, well done to authors. I hope my comments will improve the manuscript for publication. We need more data for Paralympic athletes.

Round 2

Reviewer 2 Report

Dear Authors,

Thank you for your responses. I believe that the manuscript is significantly improved. However, I have three major problems: The first deals with the experimental design and the lack of counterballanced design. This is a limitation for this study. There is a lack of time-control and effect on performance from the different conditions. Secondly, the lack of simultaneously feedback during the lifts is also a limitation. There are numerous of research evidence that participants perform better when they know and see their actual performance. I strongly believe that if these two conditions were met, then this study might have different results.

Fianlly, it is well known that during powerful lifts athletes are instructed to perform the lifts with maximum velocity during the concentric phase of movement regardless the actual movement velocity. This is another limitation of the current study. In line with the previous two issues I think that this study would be totally different.

Please, add these changes in the limitation paragraph and consider my suggestions for your future studies in Parapowerlifters since more research is needed in this sport.

Author Response

Dear reviewer

Thank you for your considerations and suggestions. We present here, marked in pink, the changes made as a result of your relevant evaluation. We will keep your observations in mind for our future studies on parapowerlifting.

Reply to the reviewer:  the changes were made to the manuscript on line 344